# Influence of Family Environment and Tobacco Addiction: A Short Report from a Post-Graduate Teaching Hospital, India

**DOI:** 10.3390/ijerph17082868

**Published:** 2020-04-21

**Authors:** Rohit Sharma, Natália Martins, Arunabh Tripathi, Pasquale Caponnetto, Neha Garg, Eugenie Nepovimova, Kamil Kuča, Pradeep Kumar Prajapati

**Affiliations:** 1Institute of Medical Sciences, Department of Rasashastra and Bhaishajya Kalpana, Faculty of Ayurveda, Banaras Hindu University, Varanasi 221005, India; 2Faculty of Medicine, University of Porto, Alameda Prof. Hernani Monteiro, 4200-319 Porto, Portugal; 3Institute for Research and Innovation in Health (i3S), University of Porto, 4200-135 Porto, Portugal; 4National Institute of Indian Medical Heritage, CCRAS, Ministry of AYUSH, Government of India, Hyderabad, Telangana 500036, India; ashoka07bhu@yahoo.in; 5Department of Education, University of Catania, 2 Ofelia, 95124 Catania, Italy; p.caponnetto@unict.it; 6Center of Excellence for the Acceleration of Harm Reduction, University of Catania, 95123 Catania, Italy; 7Institute of Medical Sciences, Department of Medicinal Chemistry, Faculty of Ayurveda, Banaras Hindu University, 221005 Varanasi, India; nehagarg@bhu.ac.in; 8Department of Chemistry, Faculty of Science, University of Hradec Králové, 50003 Hradec Králové, Czech Republic; eugenie.nepovimova@uhk.cz; 9Department of Rasashastra and Bhaishajya Kalpana, All India Institute of Ayurveda, New Delhi 110076, India; prajapati.pradeep1@gmail.com

**Keywords:** addiction, family, Jamnagar, public health, smoking, tobacco, smoking

## Abstract

*Background:* The initiation of tobacco addiction is complex, and several factors contribute to the onset of this behavior. It is presumed that the influence of family environment may pose a key factor in tobacco addiction. Tobacco-use has been highly observed in the Jamnagar district of Saurashtra region of Gujarat, India. No earlier study has focused on determining the pervasiveness of tobacco-use in families of tobacco users and non-users in this geographical area. Thus, this study aimed to assess the practice and pattern of tobacco-use (smoking and/or tobacco-chewing) in the families of tobacco-user patients. *Methods:* We studied the families of 65 tobacco-user patients (Group 1) who visited an outpatient clinic of an Ayurvedic post-graduate hospital with complaints of cough were studied and compared with age and gender-matched non-tobacco users (Group 2). The prevalence of tobacco use among the parents, siblings, and children of both groups was analyzed and compared. *Results:* The findings revealed that tobacco use among parents, siblings, and children in Group 1 was higher than Group 2 (*p* < 0.001). This meant that the problems of tobacco addiction are not always related to the individual, and therefore, tobacco-prevention strategies should focus on the entire family. *Conclusions:* These findings offer further insight into the promotion of smoking prevention interventions. Nevertheless, further research is warranted.

## 1. Introduction

Despite considerable efforts over the last decades in the fight against tobacco use, it remains the prominent cause of various preventable diseases [1,2]. Tobacco use is estimated to kill over 5 million individuals worldwide per year [3]. Moreover, the epidemic of tobacco addiction is shifting from developed to developing nations [4]. India is the second-largest consumer of tobacco which has impacted the public health significantly. India alone accounts for 1/6th of global tobacco-ailments and is likely to confront an exponential rise in tobacco-linked mortality from 1.4% (in 1990) to 13.3% (in 2020) [5].

It is imperative to determine the factors responsible for the initiation of tobacco-use to design an effective plan to control tobacco-use, thereby preventing it in the early phases. The socio-cultural factors and family environment that promote the onset of tobacco use differ from nation to nation, from developed to developing countries, region to region, and culture to culture [6]. The family has the most direct and lasting impact, not only in education and psycho-intellectual development but also in shaping values, attitudes, manners, and habits in children [7]. It is believed that tobacco intake is mostly learnt at home; and thus, effective methods for tobacco control should first target the home environment. However, a multidimensional approach should still be used in tobacco use intervention. For instance, tobacco consumption has been highly observed in the Jamnagar district of the Saurashtra region of Gujarat, India. An earlier prevalence study conducted at Jamnagar found that among tobacco users, consumption was spread as follows: only tobacco chewing: 66.2%, only smoking: 14.6%, and both smoking and tobacco chewing: 19.2% [8]. Another study found that every four out of ten residents were exposed to chewing tobacco. Around 63.9% of current tobacco chewers and 48.2% of quitters had a history of another family member consuming tobacco in some form [9]. However, no report on the role of the family environment on tobacco addiction in this geographical area is available so far. Therefore, we have attempted to determine the practice and pattern of tobacco consumption in families of tobacco users and non-users. We also look at the impact on tobacco use among siblings and children of tobacco users and non-users in this geographical area.

## 2. Materials and Methods

In India (the Jamnagar region specifically), there are myriad forms of tobacco-usage: in smoked and smokeless forms. For smoking, Bidi/Beedi (an indigenous form of thin handmade cigarette or mini-cigar) is used more commonly in India than conventional cigarettes. Bidi is prepared by rolling dried Tendu or Temburni leaf (*Diospyros melanoxylon*) with 0.15–0.25 g of sun-dried, flaked tobacco stuffed into a conical shape, and the roll is tied with a thread. Smokeless forms of tobacco-usage include oral or nasal application, where tobacco-chewing is common in forms of gutkha, khaini, or pan masala (prepared in various forms by crushed betel nut, tobacco, catechu, lime, and sweet or savory additives).

A cross-sectional study was performed in an outpatient clinic of the Ayurvedic post-graduate teaching and research hospital at Jamnagar, India. The study commenced on August 7, 2014, and included new patients that regularly visited the outpatient clinic with complaints of cough and with a history of tobacco-usage. Sixty-five patients over the next ten months were enrolled up to May 7, 2015. Informed consent was obtained as per the Helsinki declaration after offering appropriate explanations about the study and its aims. A single examiner collected the required information in the local language. The patients with a history of tobacco use were registered in Group 1. The prevalence of tobacco use among the parents, siblings, and children was noted by forming a pedigree profile of these patients. Another 65 age and gender-matched individuals that did not use tobacco in any form were enrolled in Group 2. The tobacco non-users were the individuals who had never used any form of tobacco chewing or smoking. These tobacco non-users included healthy relatives of patients visiting the outpatient’s department and the hospital staff. Healthy controls are relatives of tobacco-user cases, so it is clear that the control belongs to same characteristics as the cases [10]. In a developing country like India, relatives have the same ethnic and socio-economic conditions. They were assessed for a history of tobacco use in the family by creating pedigree profiles for these individuals. The pedigree profile provides comprehensive evidence on smoking, psycho-social factors (conflicts in the family/stress), family history of cardiovascular ailments, hypertension, and diabetes. Pedigree assessment can provide leads for early lifestyle intervention in young asymptomatic siblings [11]. The data were presented numerically (%). The prevalence of tobacco use by the blood relatives of subjects between both groups was compared using the Chi-square/Fisher exact test and analyzed using the Statistical Package for Social Sciences (SPSS) software version 22.0 (IBM Corp., Armonk, NY, USA). The sample size for each cell in comparison to Group 1 was very low, so using the odds ratio was not valid [12]. The significance level was set at 5%.

## 3. Results

There were 65 tobacco-user patients in Group 1, and 65 age and gender-matched healthy controls in Group 2. The average age was 38.6 ± 11.7 years. There were 49 males (75%) and 16 females (25%) in Group 1, and 49 males (75%) and 16 females (25%) in Group 2. Baseline characteristics are depicted in Table 1. Socio and demographic background did not differ between both groups.

In 15 tobacco-users in Group 1, both mother and father were tobacco users. Here, 56 tobacco-user patients (86.2%) had fathers with a history of tobacco use and 16 patients (24.6%) had mothers with a history of tobacco use. In contrast, among tobacco non-users, only two participants (3.1%) had fathers with a history of tobacco use, and none (0%) had mothers with a history of tobacco use. Hence the habit of tobacco-use among parents in Group 1 was higher than in Group 2 (*p* < 0.001) (Table 2).

The siblings of the 49 patients (75.4%) in Group 1 used tobacco in different forms compared to only one (1.5%) in Group 2. Furthermore, 15 children (23.1%) of patients in Group 1 had the habit of tobacco use compared to none (0%) in Group 2. Comparatively higher tobacco consumption was observed in multiple forms among children and siblings of Group 1 patients than Group 2 patients (*p* < 0.001). Table 2 shows the practice and patterns of tobacco use among both groups.

In tobacco-user patients in Group 1, 28 (43.1%) patients used bidi, 14 (21.5%) used gutkha, 15 (23.1%) consumed tobacco orally in other forms, 8 (12.3%) used cigarettes, and 11 (16.9%) were users of poly-tobaccos. 

## 4. Discussion

It is evident from the present report that tobacco addiction was prevalent among members of tobacco-user families as compared to those of tobacco non-user families. These findings support and validate recent reports, accentuating the influence of tobacco-use in the family on developing this habit. Substantial evidence proves that children with tobacco addicted parents and siblings are at a higher risk of becoming addicted due to genetic and family environmental factors [13,14,15,16,17,18]. A young adolescent who perceives that his/her parents are permissive on the use of drugs is more likely to start an addiction to substances such as tobacco or alcohol [19]. Moreover, parents addicted to smoking are more likely to grant easy access to cigarettes and have a lesser likelihood to oppose their children’s smoking. Indeed, a study explained that tobacco addiction appears to “run in the family” and the children who grow up in families with habits of tobacco usage may duplicate it in their adult behavior based on what they have observed and learnt from their family experience. Factors such as gender, ethnicity, development stage, and social environment may also affect the risk of addiction. Thus, nature and nurture, both impact an individual’s vulnerability or resistance to such drug addictions [20]. These days, family structures have turned to be more complex, witnessing a shift from the traditional nuclear family to single-parent families, stepfamilies, foster families, and multi-generational families. Thus, when a family member starts using tobacco in any form, the whole family is affected, and such an effect may vary with the family structure. In the present study, among tobacco users, 86.2% of individuals had a father with a tobacco using history. This finding agrees with a previous report showing the considerable influence of the smoking habit of a father on the current smoking habits of his child [16]. Moreover, our study found that among tobacco users, a considerable number of individuals had a mother with a tobacco using history. Over half a million infants annually are prenatally exposed to maternal smoking. Experts have affirmed that babies born to mothers who smoked when pregnant have an increased risk of nicotine addiction and a higher chance of smoking behavior in adulthood. There is a 5-fold heightened risk of adolescent drug addiction (early tobacco experimentation or nicotine dependence) in children exposed to maternal smoking during pregnancy [21,22,23,24]. The existing theory for hiked nicotine dependence in offspring of smoker mothers’ states that nicotine crosses the placental barrier, enters the bloodstream of the developing baby (thus, inducing passive nicotine inhalation), and interacts with the genomes that govern cell differentiation, permanently altering cells’ responsiveness in such a manner that increases susceptibility to tobacco addiction [25,26]. Prenatal exposure to nicotine may result in behavioral and neural consequences, and the brain of the fetus gets primed for nicotine craving, leading to substance use (particularly smoking) in adolescence [27,28,29]. In the present report, among tobacco-users, 75.4% of individuals had a history of tobacco-user sibling(s). Our findings corroborate a recent study wherein the smoking behavior of siblings showed important influences on children’s cognitive susceptibility toward smoking addiction [30].

Our findings also revealed that a paternal smoking history was commonly observed in cases of male smokers and more maternal smoking patterns in female smokers. Research evidence also suggests that gender differences also affect the influence of family/social factors on smoking uptake in children [31,32]. For instance, differential effects are reported according to gender (with a stronger effect of father’s smoking on boys and stronger influence of mother’s smoking on girls), developmental period (with a stronger effect of parental smoking before age 13 than afterwards), and residence of parents (with the effects of father’s smoking being dependent on living in the same household as the adolescent) [31,33].

The immature mind of a child learns from what s/he sees in their surroundings, and they follow the path of what the elders do and acquire their habits accordingly. The family and elder siblings have the first, most direct and lasting impact on shaping the behavior, manners, and habits of kids. If the kids see their elder smoke/chew tobacco, they become curious and seek to imitate and tend to initiate this habit at a very young age [34]. Social learning theory suggests that learning (of an attitude or habit) occurs through or from observing others smoking and is gender-specific [35,36], i.e., the son normally copies the father’s conduct or actions. At the same time, daughters tend to copy their mothers. This phenomenon may be observed in smoking, as indicated by the significant association stated between the father and adolescent smoking. In concordance with ‘social learning theory’, earlier reports have consistently revealed that smoking addiction in parents and siblings are the substantial risk factors for smoking initiation [37,38,39,40]. Smoking in parents is also linked to the sensed safety of casual smoking and temptation toward smoke in response to smoking-related prompts, for instance, seeing someone smoke [41,42]. These problems can be generally observed in nuclear families due to ineffectual nurture, abuse, or neglect as they turn out to be emotionally sensitive. Such factors can increase vulnerability and may lead to dependency on tobacco-based products or any other addictive behavior. The investigators estimated a 32% likelihood of influence of parental smoking on their child to try smoking and a 28% chance that parental smoking influenced their child to switch the smoking habit from a monthly to daily basis [43]. Figure 1 shows the proposed understanding of family influence and intergenerational transmission of tobacco addiction behavior. Thus, smoking in the family environment seems to be a significant source of vulnerability to smoking uptake among young adolescents, and parental or sibling smoking cessation might reduce the chance of susceptibility.

Extensive evidence indicates that tobacco addiction is closely and independently associated with common mental illness, like depression, anxiety, and stress. Individuals with such psychiatric disorders have been found to smoke at higher rates, smoke more cigarettes per day, smoke for longer, and have greater difficulty quitting, compared to all other smokers. This makes it imperative to manage addictions and psychological morbidities together and not independently. High comorbidity between tobacco-use (smoking and/or tobacco-chewing) and psychiatric illnesses has been extensively reported in past decades [44,45,46,47,48,49,50]. Thus, investigating this association was out of the scope of the present study and was excluded. Among social factors, peer influence also plays a significant role to encourage or deter tobacco addiction behavior in adolescents [51,52,53]. As the present study was focused on understanding the role of the family environment on tobacco-use, identifying peer group influences was excluded in this report.

As the main limitations of this study, we highlight the small sample size and the fact that the investigation only comprised of family history from patients and did not include healthy smokers. However, in this study, we found that the tobacco-use habit is not merely an individual phenomenon, but rather it has a familial occurrence. Thus, healthcare methodologies aiming families, on the whole, will tend to be more effectual while formulating plans to control tobacco use. Moreover, families should be involved in prevention programs. Investigations show that parental censure toward smoking lessens the vulnerability of an adolescent to take up smoking [54,55]. Family and community-level intervention efforts should target reinforcing parental and older siblings’ disapproval of smoking and weaken probable intergenerational influences (Figure 2). Parents or other adult family members need to do the right thing by setting good examples [56].

Moreover, parents with low educational levels should be targeted by disseminating messages appropriate to their social setting, literacy, and understanding level via anti-smoking drives or programs [57]. Thus, the following recommendations should be considered as part of a general intervention model: (a) Policy-level methods: increased taxes or levies on tobacco-based products, stringent smoke-free laws in public/workplaces, health warnings on tobacco products, prohibiting of tobacco advertising, promotions and sponsorships (in print and electronic media, especially movies), and curbing access to minors. (b) Community-level methods: school-based health awareness programs, youth advocacy and empowerment, tobacco-free homes (peer and family support), and mass media campaigns. (c) Individual-level approaches: counseling to promote cessation, nicotine replacement therapy, and pharmacological therapy [58,59,60,61,62].

## 5. Conclusions

The present study affirms that in families of tobacco-user cough patients, tobacco consumption by parents and siblings is higher as compared to families of non-tobacco users. A higher prevalence of tobacco-use amongst children of tobacco-consuming families was observed compared to non-user families. Family environment factors were found to be linked significantly with one’s cognitive susceptibility toward smoking, with the smoking behavior of parents and siblings being identified as vital influencing factors. Moreover, while some gender-based differential findings were found. However, these may not be adequate to warrant separate intervention approaches. Large-scale, well-executed studies involving different ethnic groups and covering larger population and geographical areas should be undertaken. The present findings offer further insight into the development of effective smoking prevention interventions; nevertheless, further research is still required.

## Figures and Tables

**Figure 1 ijerph-17-02868-f001:**
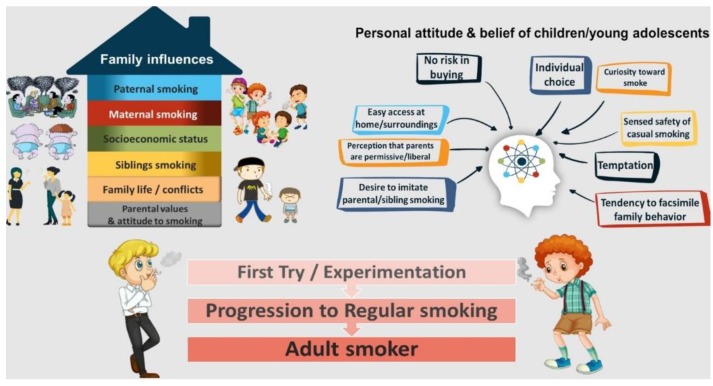
Family influence and intergenerational transmission of tobacco addiction behavior.

**Figure 2 ijerph-17-02868-f002:**
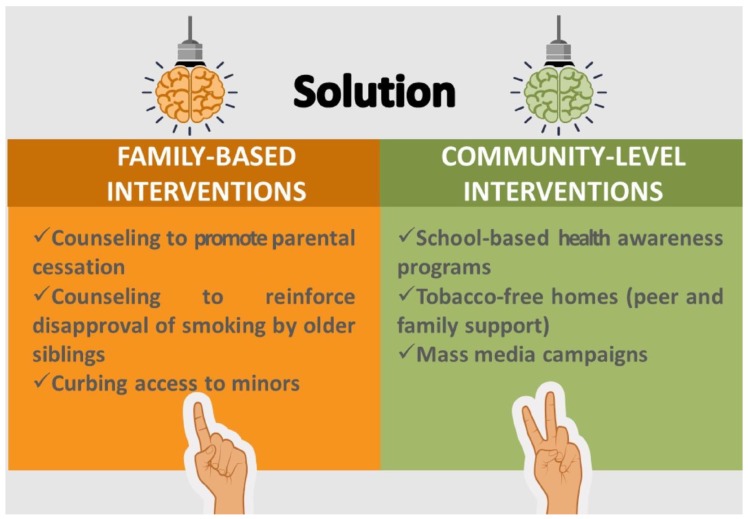
Family and community-level solutions to curb tobacco addiction.

**Table 1 ijerph-17-02868-t001:** Baseline characteristics.

Characters	Categories	Group 1 (*n* = 65)		Group 2 (*n* = 65)		
		N	%	N	%	*p* Value
Age (years)	31–40	41	63.07	41	63.07	-
	41–50	22	33.84	22	33.84	
	51–60	02	03.07	02	03.07	
Gender	Male	49	75.38	49	75.38	-
	Female	16	24.62	16	24.62	
Education	Uneducated	08	12.30	06	09.23	0.80
	Primary education	15	23.07	12	18.46	
	Secondary	19	29.23	23	35.38	
	Higher secondary	11	16.92	12	18.46	
	Graduation/Post-graduation	12	18.46	12	18.46	
Religion	Hindu	42	64.61	33	50.76	0.14
	Islamic	12	18.46	18	27.69	
	Jain	05	07.69	11	16.92	
	Others (Christian, Sikh etc.)	06	09.23	03	04.61	
Marital status	Married	63	96.92	61	93.84	0.40
	Unmarried/Widowed/ Separated	02	03.07	04	06.15	
Occupation	Business	31	47.69	33	50.76	0.64
	Govt. Employee	11	16.92	15	23.07	
	Farmer and laborer	18	27.69	14	21.53	
	Others	05	07.69	03	04.61	
Socio-economic status	Lower	14	21.53	17	26.15	0.81
	Middle	33	50.76	32	49.23	
	Upper	18	27.69	16	24.62	
Habitat	Urban	58	89.23	60	92.30	0.54
	Rural	07	10.77	05	07.70	

**Table 2 ijerph-17-02868-t002:** Tobacco usage among family members.

Parameters	Group 1 (*n* = 65)	Group 2 (*n* = 65)	*p* Value
Individuals having tobacco-user father, *n* (%)	56 (86.2)	2 (3.1)	<0.001
Individuals having tobacco-user mother, *n* (%)	16 (24.6)	0(0.0)	<0.001
Individuals having tobacco-user sibling(s), *n* (%)	49 (75.4)	1 (1.5)	<0.001

Bold: *p* < 0.001.

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
