# Peer review of "Influence of Family Environment and Tobacco Addiction: A Short Report from a Post-Graduate Teaching Hospital, India"

_ijerph, 2020, doi:10.3390/ijerph17082868_

Round 1
Reviewer 1 Report
The authors conducted cross-sectional study to assess the relationship between tobacco use and tobacco use family members. Interventions for tobacco users together with their family members might be important for tobacco control. The authors asserted family members with tobacco use might influence initiation of tobacco use among other family members, however, it was too much to say that based on the study results.
Major points
- Authors described the study design as case-control study, but the study design was cross-sectional study. If initiation of tobacco use and family tobacco use at the time of tobacco initiation was investigated, the study design would be case-control study. Because prevalence of tobacco use of the study subjects and prevalence of tobacco use of family members at the time of the study was investigated, the study was cross-sectional. The words "cases" and "controls" would mislead readers to understand the study was case-control study. Please rephrase the words "cases" and "controls".
- The results of the study did not support the conclusion. Cross-sectional study cannot prove causation. Due to small sample size and limited study area, external validity of the study results might be unclear. I think it was overstatement to say familial tobacco use cause tobacco addiction.
Minor points
- Definition of tobacco products should be declared in Materials and Methods section.
- Criteria of tobacco non-users' enrollment was unclear. More detailed description would be required to assess the comparability of two groups.
Author Response
The authors conducted cross-sectional study to assess the relationship between tobacco use and tobacco use family members. Interventions for tobacco users together with their family members might be important for tobacco control. The authors asserted family members with tobacco use might influence initiation of tobacco use among other family members, however, it was too much to say that based on the study results.
Answer: Thank you for your valuable remarks.
Major points
Authors described the study design as case-control study, but the study design was cross-sectional study. If initiation of tobacco use and family tobacco use at the time of tobacco initiation was investigated, the study design would be case-control study. Because prevalence of tobacco use of the study subjects and prevalence of tobacco use of family members at the time of the study was investigated, the study was cross-sectional. The words "cases" and "controls" would mislead readers to understand the study was case-control study. Please rephrase the words "cases" and "controls".
Answer: It is case control study but having a different angle. The question asked about their history of tobacco chewing, while their family members assessed for the same at the time of interview.
The results of the study did not support the conclusion. Cross-sectional study cannot prove causation. Due to small sample size and limited study area, external validity of the study results might be unclear. I think it was overstatement to say familial tobacco use cause tobacco addiction.
Answer: It is right that study has small sample size, but it is a matched case-control study; so in case of small sample size it is reliable because it already removes the effect of external factor in comparison of case control.
Minor points
Definition of tobacco products should be declared in Materials and Methods section.
Answer: We have incorporated your suggested points in the Materials and Methods section of the manuscript.
Criteria of tobacco non-users' enrollment was unclear. More detailed description would be required to assess the comparability of two groups.
Answer: Thank you for your valuable suggestion. Now we have incorporated this in the Materials and Methods section
Reviewer 2 Report
A useful paper for India in general and the Jamnager region specifically.
A couple of minor but important points to consider:
You occasionally refer to “tobacco abuse”. This is unfair, as tobacco is unlike narcotics which are illegal, or alcohol which is legal but can place users and innocent persons in immediate danger. As much as we might dislike it, tobacco is a legal product and these people are users of this product. They may have been introduced to it as children but are now addicted. Please don’t call them ‘abusers’, that grossly simplifies the situation and tags users with a value laden term.
Tobacco addiction is closely and independently associated with common mental illness like depression, anxiety and stress. Did you measure this among your cohorts? I’d like to see mention of this relationship in your discussion. People with mental illness, even the common types mentioned above, smoke at higher rates, smoke more cigarettes per day, smoke for longer, and have greater difficulty quitting, compared with all other smokers. There have been many papers on this topic over the past 20 years.
Also, please complete a detailed English language and grammar check.
Author Response
A useful paper for India in general and the Jamnager region specifically. A couple of minor but important points to consider:
You occasionally refer to “tobacco abuse”. This is unfair, as tobacco is unlike narcotics which are illegal, or alcohol which is legal but can place users and innocent persons in immediate danger. As much as we might dislike it, tobacco is a legal product and these people are users of this product. They may have been introduced to it as children but are now addicted. Please don’t call them ‘abusers’, that grossly simplifies the situation and tags users with a value laden term.
Answer: Thank you for your observation and valuable suggestions. We have revised the term ‘Abusers’ at suitable places of discussion section
Tobacco addiction is closely and independently associated with common mental illness like depression, anxiety and stress. Did you measure this among your cohorts? I’d like to see mention of this relationship in your discussion. People with mental illness, even the common types mentioned above, smoke at higher rates, smoke more cigarettes per day, smoke for longer, and have greater difficulty quitting, compared with all other smokers. There have been many papers on this topic over the past 20 years.
Answer: We completely agree with you, but we did not collect the data to study this relationship. It is a short report. We have now mentioned your suggested points in the discussion section.
Also, please complete a detailed English language and grammar check
Answer: Corrections are made at suitable places to improve English and grammar
Reviewer 3 Report
This is an interesting short paper largely because it draws attention to the importance of family environment on smoking/ tobacco use. From the referencing, it appears that there are other studies on this but this study reports from a context where there is less research. There are a few issues to address and these would strengthen the study especially for international readers.
- The authors mention a number of different ways in which tobacco is used (text following Table 2). In fact, smoking seems to be less used than other modes. This raises the question – Is there any difference in the addictiveness or in the related harms from the different types of tobacco usage? This could be briefly addressed in the introduction since international readrs (as myself) may not be familiar with these forms of use.
- How was ‘history of tobacco use’ and ‘did not have a habit of tobacco use’ measured? E.g. was it current use / past history of use (how long), ever used tobacco?
- I did not quite understand comments regarding some of the controls being from the same family groups as the tobacco users. How many? Perhaps explain this a bit more and consider whether it may have influenced results.
- In places the terms ‘drug abuse’ and ‘substance abuse’ are used and it is clear from one reference at least that this may be referring to substances other than tobacco. This is confusing as the paper is not discussing wider issues of substance misuse/ addiction. If the authors want to include consideration of the links between tobacco use and other substances, that would require a much bigger paper. Please consider amending this.
- While I accept the basic argument that the family environment is a fundamental context for starting to use tobacco, there is a lot of literature on the influence of peers. It would be interesting to know if this might be different in this specific region – and perhaps just say a few words about peer influence?
- While the focus of the paper is on addressing the family as a whole in trying to reduce tobacco use, the final (policy) sections clearly indicate that the authors are advocating a multi-component approach with family based intervention embedded in wider action. Again, this is a very important point and perhaps that model should be introduce briefly in the introduction.
- The paper still needs careful editing for the English.
Author Response
This is an interesting short paper largely because it draws attention to the importance of family environment on smoking/ tobacco use. From the referencing, it appears that there are other studies on this but this study reports from a context where there is less research. There are a few issues to address and these would strengthen the study especially for international readers.
Answer: Thank you for your valuable observation and constructive remarks.
The authors mention a number of different ways in which tobacco is used (text following Table 2). In fact, smoking seems to be less used than other modes. This raises the question – Is there any difference in the addictiveness or in the related harms from the different types of tobacco usage? This could be briefly addressed in the introduction since international readers (as myself) may not be familiar with these forms of use.
Answer: Smoking is the common mode of tobacco-usage in our study as Bidi (indigenous form of cigarette) and conventional cigarettes are commonly used smoking methods. This has been now detailed in Material and Method section of manuscript.
How was ‘history of tobacco use’ and ‘did not have a habit of tobacco use’ measured? E.g. was it current use / past history of use (how long), ever used tobacco?
Answer: This was a very simple question that you are practicing it or not. In India people admit about their habit only in case of when they are very habitual of that habit.
I did not quite understand comments regarding some of the controls being from the same family groups as the tobacco users. How many? Perhaps explain this a bit more and consider whether it may have influenced results.
Answer: It did not influence the result because the study considers the association with mother, father, sibling and children, not with whole family group. If it influences, it is eliminated the association between cases and tobacco history of mother, father, sibling and children. If we found no significant relationship in Table 2 then we could have explored this angle.
In places the terms ‘drug abuse’ and ‘substance abuse’ are used and it is clear from one reference at least that this may be referring to substances other than tobacco. This is confusing as the paper is not discussing wider issues of substance misuse/ addiction. If the authors want to include consideration of the links between tobacco use and other substances, that would require a much bigger paper. Please consider amending this.
Answer: Thank you for your valuable suggestions. We have made revisions at suitable places of discussion section.
While I accept the basic argument that the family environment is a fundamental context for starting to use tobacco, there is a lot of literature on the influence of peers. It would be interesting to know if this might be different in this specific region – and perhaps just say a few words.
Answer: Thank you for the suggestion. A statement on Peer influence is now mentioned in the discussion section of paper.
While the focus of the paper is on addressing the family as a whole in trying to reduce tobacco use, the final (policy) sections clearly indicate that the authors are advocating a multi-component approach with family-based intervention embedded in wider action. Again, this is a very important point and perhaps that model should be introduce briefly in the introduction.
Answer: All aspects were carefully considered in the revised version of the manuscript.
The paper still needs careful editing for the English
Answer: Corrections are made at suitable places to improve English and grammar
Round 2
Reviewer 1 Report
The term "association" should be used in case exposure and outcome have causal relationship. As the authors stated, status of tobacco use at the time of study enrollment was investigated. That is, prevalence of family tobacco use was compared between prevalent tobacco users and non-users. The results of the study meant two possible situation, study subjects' tobacco use might cause family tobacco use or family tobacco use might cause subjects' tobacco use. I thought it was resonable that elderly family members' tobacco use would cause younger family members' tobacco use. However, no data and results to support the causation was shown in the study. It was only a speculation. Usual case-control study collects an exposure before an outcome occurs, therefore, the exposure must not a result of the outcome. Subjects and their family members started tobacco use before the study enrollment, but no data can be availabe who use tobacco first. This is the reason why I did not recognize the study design as case-control study. If the authors assert the study was case-control study, show the results to support the causality of family members' tobacco use. The possibility which subjects' tobacco use caused family members' tobacco use must be denied. In addition, explain the reason why the study design was case-control study, not cross-sectional study. Due to the reason above, reliability of the "association" between subjects' tobacco use and family members' tobacco use was not high. Small sample size also supported the low reliability. However, the authors strongly asserted the results; e.g. the influential impact of family milieu in tobacco addiction is `compelling and undeniable' (line 235). The strong assertion could be made by the results from multiple large population-based cohort studies or their meta-analysis. Modest words should be used. The authors' answer to external validy was insufficient. External validity is a validity in other population. In other words, external validity is generalizability of study results. Discuss the possibility that similar results could be available when similar study was conducted in other populations. For example, the subjects of the study were elderly patients with cough in Jamnagar. Did the authors think the results were applicable to healthy elderly or young people in Jamnagar? Did it applicable to people living outside of Jamnagar, such as India, Asia or other regions?Author Response
We are very thankful to reviewer for such nice scientific review especially on design issue. We have some justification as per reviewer response.
The term "association" should be used in case exposure and outcome have causal relationship.
Answer: Here the term association was used to describe the relationship between two nominal variables. The term “Association” was not a basis to define any causal relationship. Any association is causal only if it follows certain defined criteria as mentioned in various literatures.
As the authors stated, status of tobacco use at the time of study enrollment was investigated. That is, prevalence of family tobacco use was compared between prevalent tobacco users and non-users. The results of the study meant two possible situation, study subjects' tobacco use might cause family tobacco use or family tobacco use might cause subjects' tobacco use. I thought it was reasonable that elderly family members' tobacco use would cause younger family members' tobacco use. However, no data and results to support the causation was shown in the study. It was only a speculation. Usual case-control study collects an exposure before an outcome occurs, therefore, the exposure must not a result of the outcome. Subjects and their family members started tobacco use before the study enrollment, but no data can be available who use tobacco first. This is the reason why I did not recognize the study design as case-control study. If the authors assert the study was case-control study, show the results to support the causality of family members' tobacco use. The possibility which subjects' tobacco use caused family members' tobacco use must be denied. In addition, explain the reason why the study design was case-control study, not cross-sectional study.
Answer: It is very right that the subject started tobacco’ use before the study enrolment, and that the prevalence of family tobacco use was compared between prevalent tobacco users and non-users. In addition, there are two possible ways to think here as suggested by reviewer, but the mentioned statement that “subjects' tobacco use might cause family tobacco use” is not applicable in Indian scenario, because in Indian culture, ethically a young person or adolescent cannot offer any tobacco practice to his/her elders. With regards to the study design we considered and agreed with the reviewer comment, and thus we changed the study design to cross-sectional.
Due to the reason above, reliability of the "association" between subjects' tobacco use and family members' tobacco use was not high. Small sample size also supported the low reliability. However, the authors strongly asserted the results; e.g. the influential impact of family milieu in tobacco addiction is `compelling and undeniable' (line 235). The strong assertion could be made by the results from multiple large population-based cohort studies or their meta-analysis. Modest words should be used.
Answer: Thank you for the reviewer comment. We carefully addressed all these comments, and proper changes were made in the revised version of the manuscript.
The authors' answer to external validity was insufficient. External validity is a validity in other population. In other words, external validity is generalizability of study results. Discuss the possibility that similar results could be available when similar study was conducted in other populations. For example, the subjects of the study were elderly patients with cough in Jamnagar. Did the authors think the results were applicable to healthy elderly or young people in Jamnagar? Did it applicable to people living outside of Jamnagar, such as India, Asia or other regions?
Answer: Yes, these findings may also be applicable for/to healthy elderly or young people in Jamnagar. Moreover, it can also be applicable to India and other Asian countries with a cultural background similar to that of India.
Once again thank you very much for your valuable suggestions. We are agreeing with your review and we have corrected our manuscript as per your suggestion